# Serum Oncostatin M in Ulcerative Colitis Patients and Its Relation to Disease Activity

**DOI:** 10.3390/ijms27010307

**Published:** 2025-12-27

**Authors:** Alina-Ecaterina Jucan, Georgiana-Elena Sarbu, Vasile-Claudiu Mihai, Carmen Atodiresei, Simona Juncu, Ioana-Ruxandra Mihai, Mariana Pavel-Tanasa, Daniela Constantinescu, Mihaela Dranga, Otilia Nedelciuc, Diana-Gabriela Iosep, Mihai Danciu, Smaranda Diaconescu, Georgiana-Emmanuela Gîlca-Blanariu, Andrei Mihai Andronic, Elena Toader, Vasile-Liviu Drug, Cristina Cijevschi Prelipcean, Catalina Mihai

**Affiliations:** 1Department of Gastroenterology, Grigore T. Popa University of Medicine and Pharmacy Iasi, 700115 Iasi, Romania; ghiata.alina-ecaterina@d.umfiasi.ro (A.-E.J.); mihai_vasile-claudiu@d.umfiasi.ro (V.-C.M.); carmen_atodiresei@yahoo.com (C.A.); juncu_simona-stefania@d.umfiasi.ro (S.J.); ioana-ruxandra_mihai@umfiasi.ro (I.-R.M.); mariana.pavel-tanasa@umfiasi.ro (M.P.-T.); d.constantinescu@umfiasi.ro (D.C.); mihaela_dra@yahoo.com (M.D.); otilianedelciuc@yahoo.com (O.N.); diana.iosep@umfiasi.ro (D.-G.I.); mihai.danciu@umfiasi.ro (M.D.); georgiana.gilca@gmail.com (G.-E.G.-B.); andrei.andronic@umfiasi.ro (A.M.A.); toader.elena@yahoo.com (E.T.); vasidrug@email.com (V.-L.D.); catalina.mihai@umfiasi.ro (C.M.); 2Department of Gastroenterology, Institute of Gastroenterology and Hepatology, Saint Spiridon County Hospital, 700111 Iasi, Romania; cristina.cijevschi.prelipcean@umfiasi.ro; 3Department of Radiology, Saint Spiridon” County Clinical Emergency Hospital Iasi, 700115 Iasi, Romania; 4Department of Rheumatology and Rehabilitation, Clinical Rehabilitation Hospital, 700661 Iasi, Romania; 5Laboratory of Immunology, Department of Immunology, St. Spiridon County Clinical Emergency Hospital, 700111 Iasi, Romania; 6Department of Morpho-Functional Sciences I (Morphopathology), Grigore T. Popa University of Medicine and Pharmacy, 16, Universitatii Street, 700115 Iasi, Romania; 7Department of Pediatrics, University of Medicine Titu Maiorescu, 040441 Bucharest, Romania; smaranda.diaconescu@prof.utm.ro

**Keywords:** ulcerative colitis, biomarker, oncostatin M, histological healing, endoscopic healing, disease activity

## Abstract

Ulcerative colitis (UC) is a chronic, relapsing inflammatory bowel disease; non-invasive biomarkers that accurately reflect the endoscopic and histological activity of UC require validation. Therefore, our study focused on exploring the potential of serum oncostatin M (OSM) as a biomarker for evaluating UC severity. A total of UC 89 eligible participants (≥18 years) underwent extensive clinical and paraclinical evaluation. Clinical, endoscopic, and histological activity were assessed using the partial Mayo score (pMS), the Mayo endoscopic score (MES), and the Nancy Histological Index (NHI), respectively. Serum OSM levels were determined by ELISA test and measured in pg/mL; fecal calprotectin (FC) was measured in µ/g. In our study, serum OSM was significantly associated with all four outcome measures: higher OSM levels predicted higher pMS (β = 0.471, *p* < 0.001, R^2^ = 0.222), MES (β = 0.422, *p* < 0.001, R^2^ = 0.178), NHI (β = 0.422, *p* < 0.001, R^2^ = 0.256), and FC (β = 0.431, *p* < 0.001, R^2^ = 0.186). Furthermore, ROC curve analyses demonstrated that OSM had excellent diagnostic accuracy for active disease, particularly in relation to histological inflammation (AUC = 0.967). In comparison, FC showed good but slightly lower accuracy (AUC = 0.875). Notably, OSM also outperformed FC in discriminating histological remission. Pairwise ROC curve analyses using DeLong’s test further confirmed the diagnostic accuracy of OSM, FC, and combined biomarker scores (OSM+FC) across clinical, endoscopic, and histological endpoints. The combined score PRE1 (OSM + FC based on NHI) achieved perfect discrimination (AUC = 1.000, *p* < 0.001). Composite models PRE2 and PRE3 (OSM+FC based on MES and pMS) improved diagnostic accuracy relative to OSM, confirming the value of combining OSM with FC. Although both outperformed OSM (*p* < 0.05), neither achieved a superior advantage over FC. Serum OSM is strongly associated with histological activity in UC and demonstrates superior performance compared with FC in assessing histological remission.

## 1. Introduction

Ulcerative colitis (UC) and Crohn’s disease (CD) are chronic, relapsing inflammatory bowel diseases (IBDs) of the gastrointestinal tract. Recent European Crohn’s and Colitis Organization (ECCO) consensus statements integrating clinical, biochemical, endoscopic, and histological criteria have improved the diagnostic accuracy for CD and UC [1]. Accurate prediction of UC progression and treatment response depends on understanding its dysregulated inflammatory pathways [2,3].

There are a multitude of factors that negatively influence the quality of life in patients with IBD [4]. Endoscopic examinations are associated with problems such as physical burden to the patient, cost, and risk of complications. There is a need for more accurate and non-invasive biomarkers for establishing disease activity in UC. Therefore, reliable non-invasive biomarkers that accurately mirror endoscopic and histological disease activity are needed to reduce reliance on repeated endoscopic procedures. Histological healing remains a crucial target, predictive of better outcomes and relapse prevention [5]. Thus, to avoid the need for frequent endoscopy, non-invasive biomarkers that accurately reflect the endoscopic and histological activity of UC require validation. An ideal non-invasive biomarker should be sensitive, specific, reproducible, cost-effective, and correlate reliably with disease activity and treatment response [5]. This highlights the importance of research to identify biomarkers that can predict treatment response in advance. Accordingly, our study focused on identifying a potential biomarker.

Oncostatin M (OSM) is a pro-inflammatory cytokine from the IL-6 family produced by immune cells such as T cells, monocytes, macrophages, dendritic cells, and neutrophils. It plays a vital role in various pathological processes, including inflammation in the intestines, joints, skin, and lungs [6]. It has been demonstrated that OSM and oncostatin M receptor (OSMR) are overexpressed in many IBD lesions [6]. OSM exerts potent pro-inflammatory effects in the intestinal microenvironment. It promotes activation of endothelial and stromal cell compartments, enhances leukocyte adhesion and recruitment, and contributes to tissue remodeling through fibroblast stimulation and fibrogenic pathways [7]. Elevated OSM levels are consistently detected in the blood and inflamed intestinal mucosa of patients with IBD, where they correlate with disease severity and highlight its potential utility as a biomarker of disease activity [7]. Moreover, OSM demonstrates strong diagnostic performance in distinguishing newly diagnosed UC patients from healthy controls (AUC = 0.945), particularly when measured in serum or mucosal biopsies, supporting its emerging role as a non-invasive biomarker [8].

Histologic improvement has been proposed as a potential surrogate marker for subsequent endoscopic remission, with implications for optimizing follow-up intervals and informing long-term disease management [9]. Emerging evidence suggests that histological healing is a more robust predictor of sustained remission and reduced relapse risk than endoscopic healing alone, underscoring its prognostic significance in both clinical management and the design of therapeutic trials in IBD [9,10]. Consequently, achieving combined histo-endoscopic remission is becoming an important treatment goal in UC treat-to-target strategies [11].

Accordingly, we explored the potential of serum OSM as a biomarker for evaluating UC severity across clinical, endoscopic, and histological activity. Moreover, integrating OSM with additional biomarkers may facilitate a more personalized therapeutic approach.

## 2. Results

### 2.1. Study Population

A descriptive analysis, illustrated in Table 1 and Table 2, was performed for all variables. A total of 89 patients with UC were included in the study. The mean age was 41.9 ± 14.5 years (range: 18–77 years). A male predominance was observed, with 56 males (62.9%) and 33 females (37.1%). Regarding inflammatory markers, the mean OSM level was 204.3 ± 407.0 pg/mL, ranging widely from 12.72 to 2338 pg/mL, reflecting high inter-individual variability, and the median value = 49.12 (IQR: 21.34–180.10)—Figure 1a. The FC mean was 626.9 µg/g, and the median was 202.0 (IQR: 55.5–701.5)—Figure 1b.

The mean C-reactive protein (CRP) concentration was 1.37 ± 2.44 mg/dL (range: 0.02–13.0 mg/dL), while the mean fibrinogen level was 370.9 ± 79.5 mg/dL (range: 215–568 mg/dL). The pMS averaged 5.79 ± 2.35 (range: 2–11), consistent with moderate disease activity. With respect to disease extent (Montreal classification), pancolitis (E3) was the most frequent presentation (47 patients, 52.8%), followed by left-sided colitis (E2) (36 patients, 40.4%) and proctitis (E1) (6 patients, 6.7%). Endoscopic remission was observed in 32 patients (36%), slightly exceeding the proportion in clinical remission (29 patients, 32.6%). This likely reflects the presence of persistent symptoms unrelated to active mucosal inflammation, such as functional bowel disturbances or post-inflammatory changes. By contrast, histological remission was achieved in only seven patients (7.9%), underscoring the marked discordance between clinical symptoms, endoscopic healing, and the far more stringent target of histological remission in UC. This underscores the need for reliable biomarkers that closely correlate with histological activity, enabling more accurate disease assessment and better therapeutic stratification.

### 2.2. Correlations: OSM with Disease Extent

Statistical analysis revealed a significant moderate positive Spearman correlation between OSM levels and disease extent, ρ(87) = 0.345, *p* = 0.001, indicating that higher OSM levels were associated with greater disease involvement—Figure 2a. After identifying significant associations through correlations, we conducted one-way ANOVA to examine group differences in OSM levels across disease extent (proctitis, left-sided colitis, pancolitis). ANOVA tests revealed a statistically significant effect of disease extent on OSM levels: F(2, 86) = 6.18, *p* = 0.003. Mean OSM levels were 44.01 pg/mL (SD = 38.52) in proctitis, 54.01 pg/mL (SD = 43.94) in left-sided colitis, and 339.97 pg/mL (SD = 524.83) in pancolitis.

Post hoc LSD tests indicated no significant differences between proctitis and left-sided colitis (*p* = 0.953) or between proctitis and pancolitis (*p* = 0.080). However, patients with pancolitis had significantly higher OSM levels compared with those with left-sided colitis (*p* = 0.001)—Figure 2b.

### 2.3. Correlations of OSM with Clinical Activity (pMS)

#### Partial Mayo Score

The results showed a significant moderate positive Pearson correlation between OSM levels and pMS, r(87) = 0.471, *p* < 0.001, indicating that higher OSM levels were associated with an increased disease activity—Figure 3a.

### 2.4. Correlations of OSM with Biological Markers (FC, CRP, Fibrinogen)

#### 2.4.1. Fecal Calprotectin

The Pearson correlation results demonstrated a significant positive association between OSM and FC values: r(87) = 0.431, *p* < 0.001. Furthermore, the correlation coefficient of 0.431 demonstrates a moderate positive relationship between these variables—Figure 3b.

#### 2.4.2. CRP, Fibrinogen

Conversely, OSM levels demonstrated no statistically significant Pearson correlation with either fibrinogen or CRP, suggesting that OSM may not reflect the systemic acute-phase response captured by traditional inflammatory markers. The results demonstrated that the Pearson correlation between OSM levels and fibrinogen levels was statistically non-significant, r(87) = 0.158, *p* = 0.140, suggesting no significant linear association between the two variables—Figure 3c. Similarly, OSM levels showed a statistically non-significant Pearson correlation with CRP, r(87) = 0.127, *p* = 0.237, indicating no significant linear relationship between the two variables—Figure 3d.

#### 2.4.3. Multiple Linear Regression

The multiple linear regression conducted allowed us to examine the combined and independent effects of CRP, fibrinogen, and pMS on OSM levels, controlling for the influence of each predictor within the model. The overall regression model was significant, F(3, 85) = 8.08, *p* < 0.001, explaining 22.2% of the variance in OSM levels (adjusted R^2^ = 0.19). Among the predictors, only pMS was a significant predictor of OSM levels (β = 0.46, t = 4.61, *p* < 0.001). Neither fibrinogen (β = 0.01, t = 0.12, *p* = 0.91) nor CRP (β = 0.01, t = 0.09, *p* = 0.93) were statistically significant predictors in the model.

### 2.5. Correlations of OSM with Endoscopic Activity

The results of the Pearson correlation demonstrated a significant positive association between OSM values and MES, r(87) = 0.422, *p* < 0.001, on a sample of 89 patients. The obtained correlation coefficient, 0.422, demonstrates a moderate positive relationship between these two variables—Figure 3e.

### 2.6. Correlations of OSM with Histological Activity

The Pearson correlation results revealed a significant positive association between OSM values and NHI, r(87) = 0.506, *p* < 0.001, on a sample of 89 patients. This time, the correlation coefficient of 0.506 indicates a strong positive relationship between the two variables—Figure 3f.

### 2.7. Combined Biomarkers

ROC curve analyses were performed to assess the diagnostic accuracy of OSM, FC, and their combined biomarker score (PRE 1, PRE 2, PRE3) in distinguishing active disease from remission, based on the NHI, MES, and pMS. The combined biomarker score (PRE = OSM + FC) variable was created using binary logistic regression to combine OSM and FC values into a single predictive score, allowing the combined marker to be evaluated in the ROC analysis alongside the individual biomarkers.

#### 2.7.1. PRE1 Based on NHI

OSM demonstrated excellent diagnostic performance, with an AUC of 0.967 (SE = 0.027, 95% CI [0.913, 1.000], *p* < 0.001). The optimal cut-off point for OSM was 21.75 pg/mL, which provided a sensitivity of 80.5% and a specificity of 100%—Figure 4a. On the other hand, FC showed good but lower accuracy, with an AUC of 0.875 (SE = 0.051, 95% CI [0.776, 0.974], *p* = 0.001). The best cut-off for FC was 137.5 µg/g, resulting in a sensitivity of 63.4% and a specificity of 100%. The PRE 1 score achieved perfect discrimination, with an AUC of 1.000 (SE = 0.000, 95% CI [1.000, 1.000], *p* < 0.001)—Figure 4b. The optimal cut-off for PRE 1 was 0.50, which yielded both 100% sensitivity and 100% specificity, reflecting a perfect separation between remission and active disease groups—Figure 4c. Cut-off points are meaningful for PRE 1 as a classification threshold in ROC analysis, but the values represent predicted probabilities or risk scores from the combined model, not raw biomarker levels.

Pairwise comparisons of the ROC curves using DeLong’s test confirmed that OSM performed significantly better than FC (z = 2.42, *p* =0.015). Similarly, PRE 1 significantly outperformed FC (z = 2.95, *p* = 0.003). However, the difference between OSM and PRE 1 was not statistically significant (z = 1.56, *p* = 0.120), indicating that combining the two biomarkers does not provide a significant improvement over OSM alone—Figure 4d.

#### 2.7.2. PRE2 Based on MES

OSM demonstrated good diagnostic accuracy in differentiating active endoscopic disease from remission, with an AUC of 0.756 (SE = 0.050, 95% CI [0.657, 0.855], *p* < 0.001). The optimal cut-off point was 22.36, providing a sensitivity of 84.2% and a specificity of 46.9%—Figure 5a. Regarding FC, the results of our statistical analysis showed slightly higher AUC compared to OSM, at 0.838 (SE = 0.043, 95% CI [0.754, 0.923], *p* < 0.001). The optimal cut-off value of 178.5 resulted in 77.2% sensitivity and 87.5% specificity—Figure 5b. The PRE2 score demonstrated good diagnostic accuracy in predicting endoscopic activity, with an AUC of 0.873 (SE = 0.037, 95% CI [0.800, 0.946], *p* < 0.001). The optimal cut-off value was approximately 0.456, yielding a sensitivity of 84.2% and a specificity of 81.2%—Figure 5c. These results indicate that PRE 2 can reliably distinguish active disease from remission. However, it should be mentioned that cut-off points are meaningful for PRE 2 as a classification threshold in ROC analysis, but the values represent predicted probabilities or risk scores from the combined model, not raw biomarker levels.

Pairwise comparisons of the ROC curves using DeLong’s test revealed that PRE 2 performed significantly better than OSM (z = 2.14, *p* = 0.032). However, the difference between FC and OSM was not statistically significant (z = 1.47, *p* = 0.141), and there was no significant difference between PRE 2 and FC (z = 0.76, *p* = 0.449)—Figure 5d.

#### 2.7.3. PRE3 Based on pMS

In the analysis, a pMS of 0 to 2 was considered clinical remission, whereas a score of 3 to 12 was classified as active disease. OSM demonstrated good diagnostic performance, with an AUC of 0.738 (SE = 0.052, 95% CI [0.636, 0.840], *p* < 0.001). The optimal cut-off point for OSM was 23.47 pg/mL, which provided a sensitivity of 81.7% and a specificity of 48.3%—Figure 6a. On the other hand, FC showed better diagnostic accuracy than OSM, with an AUC of 0.843 (SE = 0.041, 95% CI [0.762, 0.923], *p* < 0.001). The best cut-off for FC was 190 µg/g, yielding a sensitivity of 73.3% and a specificity of 89.7%—Figure 6b. Although FC demonstrated slightly higher diagnostic accuracy than OSM, the current analysis does not provide conclusive evidence of its superiority. The optimal cut-off for PRE3 was 0.52, which yielded a sensitivity of 80% and a specificity of 86.2%—Figure 6c. This combined score outperformed each biomarker alone.

Pairwise comparisons using DeLong’s test showed that PRE 3 performed significantly better than OSM alone (z = 2.08, *p* = 0.037). The differences between OSM and FC (z = 1.59, *p* = 0.11) and between FC and PRE 3 (z = 0.51, *p* = 0.61) were not statistically significant—Figure 6d.

## 3. Discussion

The need for a personalized approach for patients with UC is high. In this study, we investigated serum OSM levels from UC patients and provided further evidence for its potential as a marker of disease extension and severity (endoscopic and histological healing).

### 3.1. Serum OSM Correlates with Disease Healing in UC

In our study, simple linear regression analyses showed that serum OSM was significantly associated with all four outcome measures: higher OSM levels predicted higher pMS (β = 0.471, *p* < 0.001, R^2^ = 0.222), MES (β = 0.422, *p* < 0.001, R^2^ = 0.178), NHI (β = 0.422, *p* < 0.001, R^2^ = 0.256), and FC (β = 0.431, *p* < 0.001, R^2^ = 0.186). These findings confirm that OSM is strongly linked to both clinical and endoscopic activity, as well as histological inflammation and fecal biomarkers. Furthermore, serum OSM levels were markedly higher in patients with pancolitis (mean 339.97 pg/mL) compared with those with left-sided colitis or proctitis (means 44–54 pg/mL).

FC is a common marker of intestinal inflammation but lacks specificity for histological activity [12], whereas OSM reflects inflammation and contributes to tissue remodeling [13], showing stronger correlations with histological severity than CRP or FC [14,15]. Consistent with these findings, Cao et al. [16] also demonstrated that serum OSM is elevated in patients without mucosal healing, while Yang et al. [17] highlighted stronger correlations of OSM with mucosal and histological inflammation than with biochemical markers.

Overall, our findings suggest that serum OSM correlates strongly with UC disease extent and severity, with markedly elevated concentrations in patients with extensive disease, as well as in those with severe disease activity.

In addition, the multiple linear regression was conducted to assess the independent contributions of pMS, fibrinogen, and CRP to serum OSM levels. The model was significant (F(3, 85) = 8.08, *p* < 0.001) and explained 22.2% of the variance in OSM (adjusted R^2^ = 0.19). Notably, pMS was the only significant predictor (β = 0.46, *p* < 0.001), whereas fibrinogen and CRP did not significantly contribute to the model. This indicates that serum OSM levels are primarily driven by clinical disease activity rather than systemic inflammatory markers, suggesting that OSM may more closely reflect mucosal or disease-specific inflammation rather than generalized acute-phase response.

In our study, ROC curve analyses demonstrated that OSM had excellent diagnostic accuracy for active disease, particularly in relation to histological inflammation (AUC = 0.967), whereas FC showed good but slightly lower accuracy (AUC = 0.875). Importantly, OSM outperformed FC in assessing histological remission, as reflected by both stronger correlations with the NHI and higher diagnostic accuracy in ROC analysis. While FC remains a valuable non-invasive biomarker, OSM appears to more closely reflect microscopic inflammation, which may explain its superior performance relative to histological endpoints. The addition of FC to OSM did not significantly improve diagnostic accuracy, underscoring the role of OSM as the more sensitive biomarker of histological disease activity.

The combined score PRE1 (OSM + FC based on NHI) achieved perfect discrimination (AUC = 1.000); however, this likely reflects model overfitting and highlights the need for validation in independent cohorts. Additional composite models (PRE2–PRE3 based on MES and pMS, respectively) consistently improved diagnostic performance compared with OSM alone, supporting the value of combining OSM with FC. Specifically, PRE2 and PRE3 significantly outperformed OSM (*p* < 0.05), but neither demonstrated superiority over FC. Pairwise ROC curve analyses using DeLong’s test further confirmed the diagnostic accuracy of OSM, FC, and combined biomarker scores (PRE1–PRE3) across clinical, endoscopic, and histological endpoints, with OSM consistently showing the strongest associations with histological inflammation. These findings suggest that OSM may serve as a promising biomarker for identifying histological remission, a clinically meaningful treatment target increasingly recognized in IBD management. Taken together, these results support the potential utility of OSM as a complementary biomarker in UC, particularly in identifying patients with persistent histological activity despite biochemical improvement.

A major strength of this study lies in its comprehensive evaluation of OSM, both individually and in combination with FC, across clinical, endoscopic, histological, and biochemical outcomes, using robust statistical methods including ROC analysis and DeLong’s test. A key strength is the novel evidence that OSM may outperform FC in assessing histological remission, an increasingly recognized therapeutic target in UC. The exploration of combined biomarker models (PRE1–PRE3) also advances the field toward more integrated approaches to disease assessment.

### 3.2. Study Limitations

Despite these strengths, several limitations should be acknowledged. This study did not include a healthy control group or patients with other inflammatory conditions, as it focused on correlations between serum OSM and clinical, biochemical, endoscopic, and histological disease activity in UC. The study was not designed to evaluate diagnostic performance or treatment response. Subsequent studies are needed to assess OSM in comparative populations and to explore its utility for both treatment monitoring and diagnostic purposes.

The relatively modest, single-center cohort may restrict the generalizability of the results and increase the risk of statistical error. The perfect discrimination observed for PRE1 likely reflects model overfitting, emphasizing the need for validation in independent datasets. Our findings require validation in larger, independent studies and the establishment of standardized methodologies to define quantitative cut-offs for serum OSM before it can be implemented in routine clinical practice. Also, research focusing on OSM in other biological samples (e.g., tissue, saliva, or feces) should be considered. Moreover, the prognostic value of serum OSM warrants further investigation.

### 3.3. Clinical Implications

Our data support incorporating histologic evaluation alongside clinic and endoscopic assessment to achieve deeper remission in UC. The findings of our study suggest that serum OSM could serve as a valuable biomarker in the personalized management of UC. OSM measurement may help identify patients with histologic active disease, highlighting its role as a biomarker of mucosal inflammation, monitoring histological activity, and guide treatment decisions. Its strong correlation with microscopic inflammation positions OSM as a potential tool for assessing histological healing, which is increasingly recognized as a key treatment target in UC management. Also, its ability to more accurately identify patients with persistent histological inflammation could optimize therapeutic decision-making, potentially reducing unnecessary escalation of therapy or ineffective treatments. In particular, OSM could serve as a non-invasive biomarker to help identify patients with persistent microscopic inflammation, potentially reducing the frequency of endoscopic assessments in treat-to-target strategies.

A comparative evaluation of biomarkers against histological and endoscopic reference standards provides essential guidance for optimizing daily clinical practice. Traditional systemic markers such as CRP and fibrinogen are non-specific markers of inflammation but lack the precision needed to accurately reflect mucosal activity or confirm deep healing. In contrast, biomarkers that more directly represent local tissue inflammation—particularly OSM—demonstrate superior correlations with histological activity and histologic healing, making it valuable for identifying persistent microscopic disease. FC remains a strong non-invasive indicator of mucosal inflammation, but OSM stands out for its ability to detect subclinical, treatment-resistant inflammation and to better predict true tissue-level remission. Integrating these more sensitive biomarkers into routine assessment, alongside clinical evaluation and endoscopic findings, supports a more accurate and individualized approach to disease management.

### 3.4. Cost-Effectiveness Considerations

Regarding cost-effectiveness and feasibility, while OSM measurement would involve additional laboratory costs compared to established markers such as FC, its ability to more accurately reflect mucosal inflammation could optimize therapy selection and reduce unnecessary escalation, potentially offsetting costs over time. By guiding more precise interventions, OSM testing could contribute to improved long-term outcomes and cost savings within UC management pathways.

Taken together, our findings and those of previous studies support the role of serum OSM as a biomarker closely linked to disease extent and activity (endoscopic and histological healing) in UC. OSM could be integrated into existing clinical practice alongside established biomarkers and endoscopic assessments. Combined with clinical scoring systems and fecal biomarkers, OSM may help clinicians identify patients who require intensification of therapy or closer follow-up. Validation in larger, independent cohorts will be important before broad clinical implementation, but our findings provide a strong rationale for its inclusion in future treatment algorithms.

## 4. Materials and Methods

### 4.1. Study Design and Ethical Approval

This study is a prospective investigation conducted at a tertiary referral hospital in northeastern Romania. The study group consisted of patients who were enrolled consecutively. Patient data and blood samples were collected and analyzed from April 2024 to December 2024. All UC patients had a confirmed diagnosis based on clinical symptoms, laboratory tests, imaging, endoscopy, and histopathology.

The present study was approved by the Ethics Committees of Grigore T. Popa University of Medicine and Pharmacy Iasi, Romania, and the “Saint Spiridon” County Clinical Emergency Hospital Iasi, Romania. The study was conducted in accordance with the ethical guidelines outlined in the Declaration of Helsinki. All participants provided written informed consent.

A total of 89 eligible participants (≥18 years), both male and female, had a confirmed diagnosis of UC based on clinical, endoscopic, and histological criteria in accordance with the ECCO guidelines. Demographic information, clinical assessments, and medical histories were recorded for all patients. Exclusion criteria included indeterminate colitis, bacterial superinfection, colorectal cancer or prior malignancy, and other systemic immune-mediated diseases (e.g., rheumatoid arthritis). All patients underwent extensive clinical and paraclinical evaluation. For all included patients, we collected data on the diagnosis date, disease extension (Montreal classification), clinical assessment (partial Mayo score), endoscopic activity (Mayo endoscopic score), and histological activity (Nancy Histological Index), as well as current treatments. Serum samples were collected from all UC patients, whether in remission or experiencing active disease.

Blood samples were collected from all patients, and the following tests were performed in our center: complete blood count, C-reactive protein (CRP), fibrinogen, and serum OSM. Fecal calprotectin (FC) was determined by ELISA test and was collected before bowel preparation. Clinical, endoscopic, and histologic activity were assessed as follows: disease extension using Montreal classification: E1 proctitis, E2 left side colitis, E3 pancolitis; clinical activity using the partial Mayo score (pMS—total score 12): remission ≤ 2; endoscopic activity using the Mayo endoscopic score (MES 0–3) according to vascular pattern, bleeding, and ulcers: remission 0–1, mild 2, moderate-to-severe 3; histologic activity using the Nancy Histological Index (NHI 0–4): remission 0, ≥1 active disease. Endoscopy and biopsy were performed within a maximum of 1 month from the time of sample collection.

Blood samples for OSM were collected simultaneously with the routine blood tests obtained during the patients’ standard evaluations. Blood for OSM was collected in serum (no-gel or K2-EDTA) tubes, processed within 1–2 h, and plasma aliquots were extracted via centrifugation (1000× *g*, 20 min, 18 °C). The resulting plasma aliquots in cryovials were stored at −80 °C for periods ranging from 3 to 11 months, depending on the collection date, to avoid freeze–thaw cycles until batch analysis. This storage duration is well within the established stability window for cytokines, as prior studies have demonstrated that serum and plasma cytokines remain stable for at least 2 years when kept at −80 °C [18]. A commercial enzyme-linked immunosorbent assay (ELISA) kit—Human OSM (oncostatin M) ELISA Kit (E-EL-H2247)—was used for the colorimetric detection and quantification of human OSM as per the manufacturer’s protocol, with a sensitivity of 9.38 pg/mL. OSM serum levels were measured in pg/mL.

### 4.2. Statistical Analysis

IBM SPSS Statistics 26.0 software was used for the statistical processing. Continuous variables were presented as means ± standard deviation (SD) depending on distribution. Considering that pMS, CRP, and fibrinogen are numerical variables, Pearson correlation coefficients were calculated to assess their linear relationships with OSM levels. On the other hand, for the categorical clinical variables, disease extent with three levels (proctitis, left-sided colitis, and pancolitis) of Spearman’s correlations were used instead. One-way ANOVA was used to examine group differences in OSM levels across disease extent (E1, E2, E3). The multiple linear regression was conducted to examine the combined and independent effects of CRP, fibrinogen, and pMS on OSM levels.

Receiver operating characteristic (ROC) curve analysis was performed, and the area under the curve (AUROC) was calculated to determine optimal cut-off values for clinical, endoscopic, and histological remission using non-invasive diagnostic tools. Moreover, ROC curve analyses were performed to assess the diagnostic accuracy of OSM, FC, and their combined biomarker score (OSM + FC) in distinguishing active disease from remission, based on the histological, endoscopic, and clinical activity. The combined biomarker score (PRE1 = OSM + FC based on NHI, PRE2 = OSM + FC based on MES, PRE3 = OSM+FC based on pMS) variable was created using binary logistic regression to combine OSM and FC values into a single predictive score, allowing the combined marker to be evaluated in the ROC analysis alongside the individual biomarkers. Figure 3d, Figure 4d, and Figure 5d were created using Artificial Intelligence. For all statistical tests, *p*-values ≤ 0.05 were considered statistically significant.

## 5. Conclusions

In conclusion, our findings demonstrated a robust correlation between OSM and UC. Serum OSM is strongly associated with clinical, endoscopic, and histological activity in UC and demonstrates superior performance compared with FC in assessing histological remission. OSM concentration can be used as an effective, non-invasive marker for UC management, particularly valuable for identifying histological remission, a clinically meaningful treatment target. Future research should focus on validating OSM as a biomarker in larger, longitudinal cohorts to confirm its utility for monitoring disease activity and predicting treatment response. Integration of OSM into multimodal biomarker panels could advance precision medicine approaches, enabling more accurate, non-invasive monitoring and better targeting of histological remission as a treatment goal.

## Figures and Tables

**Figure 1 ijms-27-00307-f001:**
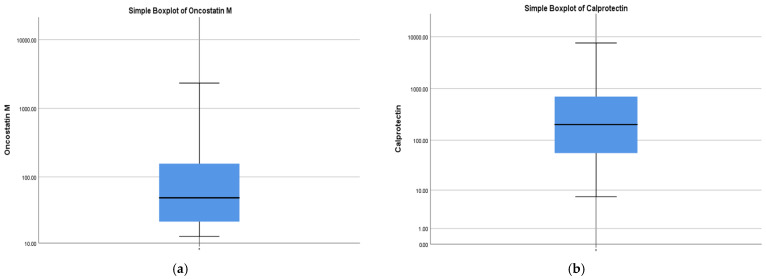
(**a**). Distribution of oncostatin M (OSM) levels across the study cohort. The Y-axis uses a logarithmic scale for enhanced visualization of the data spread. The median value is 49.12 (IQR: 21.34–180.10). (**b**). Distribution of Fecal Calprotectin (FC) levels across the study cohort. The Y-axis uses a logarithmic scale for enhanced visualization of the data spread. The median value is 202.0 (IQR: 55.5–701.5).

**Figure 2 ijms-27-00307-f002:**
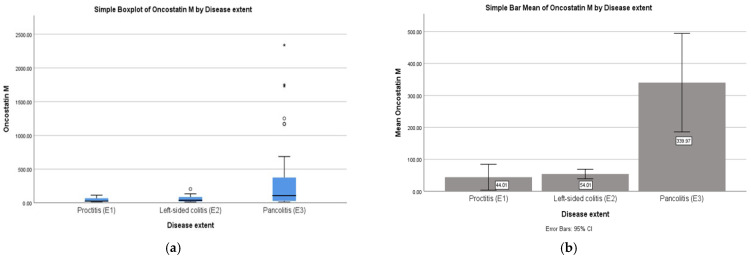
(**a**). Boxplot showing the distribution of OSM levels across disease extent categories (E1, E2, E3) in patients with UC (N = 89). A significant moderate positive Spearman’s correlation was observed (rho = 0.345, *p* = 0.001). Higher OSM levels were observed in patients with more extensive disease involvement. * denots mild outliers (>1.5-3xIQR beyond the box) and ** denots extreme outliers (>3xIQR beyond the box) (**b**). Mean OSM levels by disease extent groups: proctitis (E1), left-sided colitis (E2), and pancolitis (E3). Error bars represent standard deviations. A one-way ANOVA showed significant differences between groups, with post hoc LSD tests indicating a significant difference between left-sided colitis and pancolitis groups (**b**).

**Figure 3 ijms-27-00307-f003:**
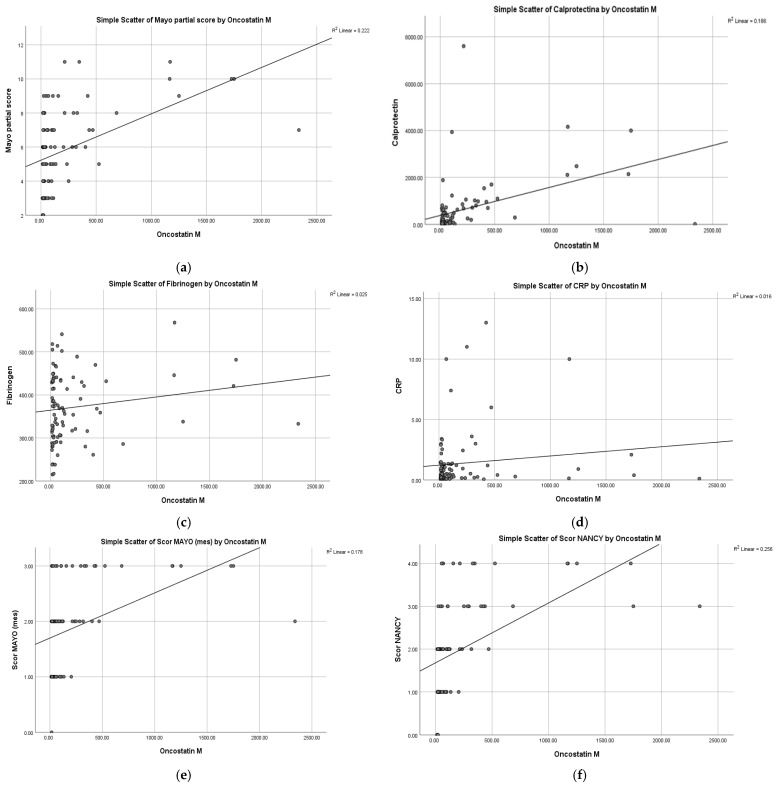
(**a**). Scatterplot showing the Pearson correlation between OSM levels and pMS in patients with UC (N = 89). The plot includes a regression line and its 95% confidence interval. Pearson’s r = 0.471, *p* < 0.001, indicating a significant moderate positive correlation. (**b**). The scatterplot illustrates the relationship between OSM values and FC levels. A moderate positive correlation is observed between the two variables (r = 0.431, *p* < 0.001, N = 89). The plotted line represents the linear regression line, indicating the overall trend of the data. (**c**). Scatterplot showing the Pearson correlation between OSM levels and fibrinogen in patients with UC (N = 89). A linear regression line with 95% confidence interval is shown. Pearson’s r = 0.158, *p* = 0.140, indicating a non-significant correlation. (**d**). Scatterplot showing the Pearson correlation between OSM levels and CRP in patients with UC (N = 89). A linear regression line with 95% confidence interval is shown. Pearson’s r = 0.127, *p* = 0.237, indicating a non-significant correlation. (**e**). Scatter plot illustrating the relationship between OSM values and MAYO score (mes). A moderate positive correlation can be observed between the two variables (r = 0.422, *p* < 0.001, N = 89). The plotted line represents the linear regression line, indicating the general trend of the data. (**f**). Scatter plot illustrating the relationship between OSM values and Nancy Score. A strong positive correlation is observed between the two variables (r = 0.506, *p* < 0.001, N = 89). The plotted line represents the linear regression line indicating the general trend of the data.

**Figure 4 ijms-27-00307-f004:**
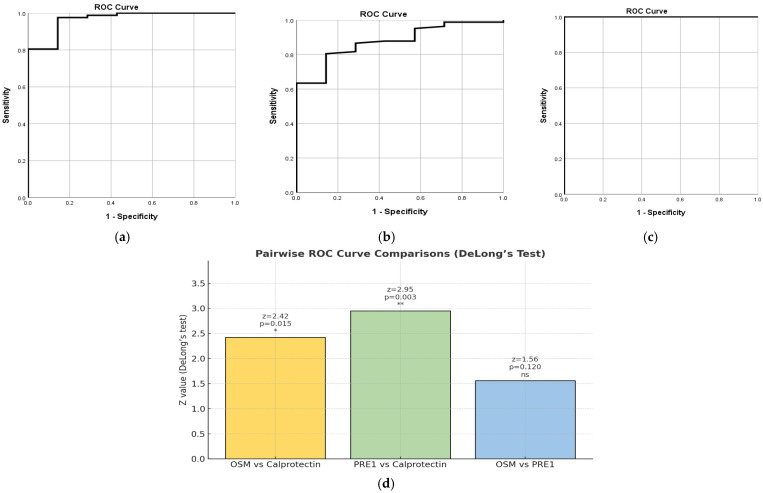
ROC Curve for OSM. The ROC curve shows the diagnostic performance of OSM in distinguishing active disease from remission based on the NHI. The area under the curve (AUC) was 0.967 (95% CI [0.913, 1.000], *p* < 0.001), indicating excellent diagnostic accuracy (**a**). ROC Curve for FC. The ROC curve displays the diagnostic accuracy of FC for differentiating active disease from remission according to the NHI. The AUC was 0.875 (95% CI [0.776, 0.974], *p* = 0.001), indicating good diagnostic performance (**b**). ROC Curve for the Combined Biomarker Score (PRE 1). This ROC curve illustrates the performance of the PRE 1 variable, created by combining OSM and FC using binary logistic regression. PRE 1 achieved perfect diagnostic discrimination with an AUC of 1.000 (95% CI [1.000, 1.000], *p* < 0.001 (**c**). Pairwise comparisons of the ROC curves using DeLong’s test (**d**). * indicates *p* < 0.05; ** indicates *p* < 0.01 based on DeLong’s test. ‘ns’ denotes not statistically significant.

**Figure 5 ijms-27-00307-f005:**
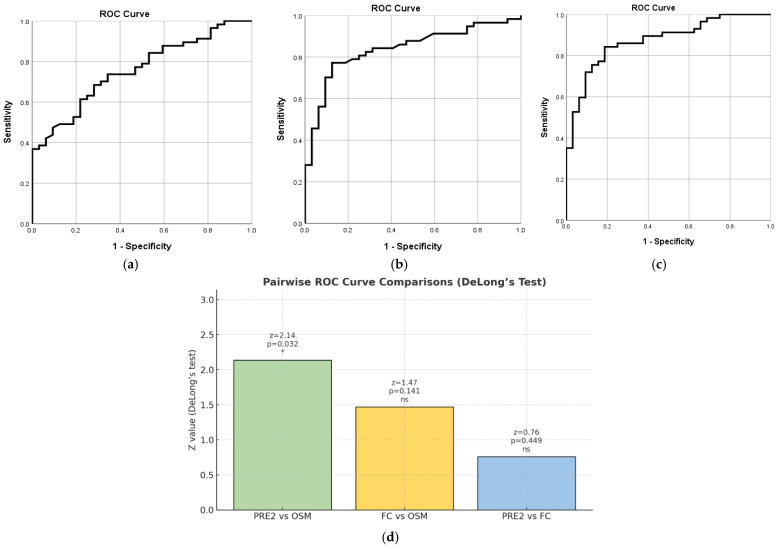
ROC Curve for OSM. The ROC curve illustrates OSM’s performance in detecting endoscopic disease activity based on the MES. AUC = 0.756 (95% CI [0.657, 0.855], *p* < 0.001) (**a**). ROC Curve for FC. FC’s performance in predicting Mayo-defined endoscopic activity is shown. AUC = 0.838 (95% CI [0.754, 0.923], *p* < 0.001) (**b**). ROC Curve for PRE 2. ROC analysis of PRE 2 (combined OSM and FC score) yielded an AUC of 0.873 (95% CI [0.800, 0.946], *p* < 0.001) (**c**). Pairwise comparisons of the ROC curves using DeLong’s test (**d**). An asterisk (*) indicates a statistically significant difference between ROC curves at *p* < 0.05; “ns” denotes not significant.

**Figure 6 ijms-27-00307-f006:**
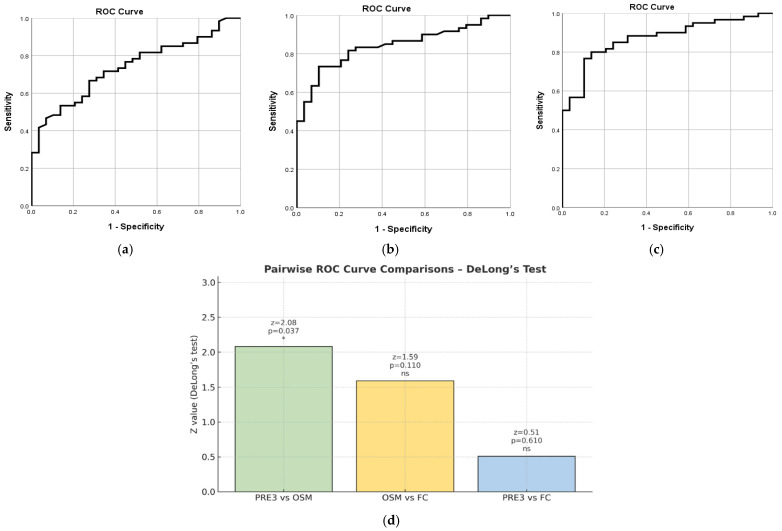
ROC Curve for OSM. The ROC curve shows the diagnostic performance of OSM in distinguishing active disease from remission based on the pMS. The AUC was 0.738 (95% CI [0.636, 0.840], *p* < 0.001), indicating good diagnostic accuracy. The optimal cut-off value was 23.47, with a sensitivity of 81.7% and specificity of 48.3% (**a**). ROC Curve for FC. The ROC curve displays the diagnostic accuracy of FC for differentiating active disease from remission according to the pMS. The AUC was 0.843 (95% CI [0.762, 0.923], *p* < 0.001), indicating strong diagnostic performance. The best cut-off point was 190, with a sensitivity of 73.3% and specificity of 89.7% (**b**). ROC Curve for PRE3. This ROC curve illustrates the performance of the PRE3 variable, created by combining OSM and FC using binary logistic regression. PRE3 achieved excellent diagnostic discrimination with an AUC of 0.871 (95% CI [0.798, 0.945], *p* < 0.001). The optimal cut-off was 0.52, yielding 80% sensitivity and 86.2% specificity (**c**). Pairwise comparisons using DeLong’s test (**d**). An asterisk (*) indicates a statistically significant difference between ROC curves at *p* < 0.05; “ns” denotes not significant.

**Table 1 ijms-27-00307-t001:** Descriptive analysis regarding gender, disease extension, and disease activity; E = extension; pMS = partial Mayo score.

Variable	Count	Percentage %
Male	56	62.9
Female	33	37.1
Proctitis (E1)	6	6.7
Left-sided colitis (E2)	36	40.4
Pancolitis (E3)	47	52.8
Clinical remission (pMS)	29	32.6
Active clinical disease (pMS)	60	67.4
Endoscopic remission	32	36
Active endoscopic disease	57	64
Histological remission	7	7.9
Active histological disease	82	92.1

**Table 2 ijms-27-00307-t002:** Descriptive analysis regarding socio-demographic, clinic, endoscopic, and laboratory characteristics; OSM= oncostatin M; pMS = partial Mayo score; MES = Mayo endoscopic score; NHI = Nancy Histological Index; FC = Fecal Calprotectin.

Variable	Median (IQR)	Mean	Standard Deviation	Minimum	Maximum
Age (years)	42.0 (30.5–53.5)	41.88	14.5	18	77
OSM (pg/mL)	49.12 (21.34–180.1)	204.3	407	12.72	2338
CRP (mg/L)	0.44 (0.20–1.30)	13.7	24.4	0.2	130
Fibrinogen (mg/dL)	368.0 (306.5–433.0)	370.96	79.46	215	568
pMS	6.0 (4.0–7.0)	5.79	2.35	2	11
MES	2.0 (1.0–2.0)	1.86	0.78	0	3
NHI	2.0 (1.0–3.0)	1.96	1.12	0	4
FC (µg/g)	202.0 (55.5–701.5)	626.9	1124	7.19	7600

## Data Availability

The original contributions presented in this study are included in the article. Further inquiries can be directed to the corresponding author.

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
