# Peer review of "Serum Oncostatin M in Ulcerative Colitis Patients and Its Relation to Disease Activity"

_ijms, 2025, doi:10.3390/ijms27010307_

Round 1

Reviewer 1 Report

Comments and Suggestions for Authors

A comparison of how each biomarker correlates with histological or endoscopic activity and remission is considered beneficial for improving the level of daily clinical practice. However, we respectfully request that you reconsider and revise the following points:

・I think that the definisions of PRE1,2,3 are unclear. Please ensure that these terms are clearly defined in the 'Methods' section.

・L270: From a statistical perspective, can you definitively conclude that FC exhibits superior diagnostic capability compared to OSM in distinguishing endoscopic remission? Could you please review the wording and consider alternative options?

Author Response

Dear Reviewer,

Thank you for your valuable time, for the positive feedback, and useful contribution on the manuscript ijms-4017391 entitled “ Serum Oncostatin M in Ulcerative Colitis Patients and Its Relation to Disease Activity”. Your suggestions and recommendations have been taken into consideration. We much appreciate their input, which helped improve our manuscript.

Please find below a detailed point-by-point reply to the comments.

Sincerely,

--

Alina-Ecaterina Jucan, MD, PhD

Grigore T. Popa University of Medicine and Pharmacy Iasi, Romania

Institute of Gastroenterology and Hepatology Iasi, Romania

Q1. A comparison of how each biomarker correlates with histological or endoscopic activity and remission is considered beneficial for improving the level of daily clinical practice.

Answer 1. To improve the article, following the reviewer’s recommendations, we have added the paragraph: ”A comparative evaluation of biomarkers against histological and endoscopic reference standards provides essential guidance for optimizing daily clinical practice. Traditional systemic markers such as CRP and fibrinogen are nonspecific markers of inflammation but lack the precision needed to accurately reflect mucosal activity or confirm deep healing. In contrast, biomarkers that more directly represent local tissue inflammation—particularly OSM—demonstrate superior correlations with histological activity and histologic healing, making it valuable for identifying persistent microscopic disease. FC remains a strong noninvasive indicator of mucosal inflammation, but OSM stands out for its ability to detect subclinical, treatment-resistant inflammation and to better predict true tissue-level remission. Integrating these more sensitive biomarkers into routine assessment, alongside clinical evaluation and endoscopic findings, supports a more accurate and individualized approach to disease management.”

However, we respectfully request that you reconsider and revise the following points:

Q2. I think that the definitions of PRE1,2,3 are unclear. Please ensure that these terms are clearly defined in the 'Methods' section.

Answer 2. We have revised this section to make the understanding of the combined score clearer: ”The combined biomarker score (PRE1= OSM+FC based on NHI, PRE2=OSM+FC based on MES, PRE3= OSM+FC based on pMS) variable was created using binary logistic re-gression to combine OSM and FC values into a single predictive score, allowing the combined marker to be evaluated in the ROC analysis alongside the individual biomarkers.”

Q3. L270: From a statistical perspective, can you definitively conclude that FC exhibits superior diagnostic capability compared to OSM in distinguishing endoscopic remission? Could you please review the wording and consider alternative options?

Answer 3.  While the reviewer’s comment refers to endoscopic remission, our study describes in L270 the combined score (OSM+FC) based on clinical activity.

If the reviewer refers to ”It should be mentioned that cutoff points are meaningful for PRE 2 as a classification threshold in ROC analysis, but the values represent predicted probabilities or risk scores from the combined model, not raw biomarker levels.”, this phrase was deleted.

And this phrase was added:”Although FC demonstrated slightly higher diagnostic accuracy than OSM, the current analysis does not provide conclusive evidence of its superiority.”

Reviewer 2 Report

Comments and Suggestions for Authors

This prospective study addresses an essential clinical need in the management of inflammatory bowel disease (IBD): the identification of non-invasive biomarkers that accurately reflect disease activity in ulcerative colitis (UC). The research is timely and relevant, as current clinical practice relies heavily on invasive endoscopic procedures for disease monitoring.

Key contributions of this study include:

  • Comprehensive evaluation of OSM across multiple disease activity measures (clinical, endoscopic, histological)
  • Direct comparison with fecal calprotectin (FC), an established biomarker
  • Novel evidence suggesting OSM may outperform FC in assessing histological remission
  • Exploration of combined biomarker models to enhance diagnostic accuracy
  • The study's focus on histological healing is particularly noteworthy, as this endpoint is increasingly recognized as a crucial therapeutic target in UC management and a predictor of long-term outcomes.

Areas for Improvement

  • The abstract, while informative, could benefit from more explicit numerical results in the Results section. For example, specific AUC values for key comparisons and p-values for the most important findings should be included to give readers immediate access to the study's primary quantitative outcomes.
  • The manuscript needs a clear, conventional structure (Introduction, Methods, Results, Discussion, Conclusion) that facilitates reader comprehension and meets journal standards.
  • The Results section contains extensive statistical detail that, while thorough, may overwhelm readers. Consider moving some technical statistical details to supplementary materials and focusing the main text on the most clinically relevant findings.
  • The Discussion section is relatively brief, given the complexity of the findings. It would benefit from expansion to include more detailed interpretation of the clinical implications, comparison with specific prior studies, and discussion of mechanistic insights.
  • While figures are referenced appropriately, the text could better integrate visual data with narrative interpretation. Each figure should be discussed in more detail regarding its clinical significance.
  • Phrases like "extensive or severe disease" (line 312) conflate disease extent with disease severity, which are distinct concepts in UC. Disease extent refers to anatomical involvement (E1-E3), while severity refers to inflammatory activity. These should be distinguished clearly.
  • There is no discussion of missing data or how it was handled. Were all 89 patients included in every analysis? If data were missing, what approach was used?
  • Discuss practical applications, cost-effectiveness considerations, and how OSM would integrate into current clinical practice and treatment algorithms.

Author Response

Dear Reviewer,

Thank you for your valuable time, for the positive feedback, and useful contribution on the manuscript ijms-4017391 entitled “ Serum Oncostatin M in Ulcerative Colitis Patients and Its Relation to Disease Activity”. Your suggestions and recommendations have been taken into consideration. We much appreciate their input, which helped improve our manuscript.

Please find below a detailed point-by-point reply to the comments.

Sincerely,

--

Alina-Ecaterina Jucan, MD, PhD

Grigore T. Popa University of Medicine and Pharmacy Iasi, Romania

Institute of Gastroenterology and Hepatology Iasi, Romania

Q1. The abstract, while informative, could benefit from more explicit numerical results in the Results section. For example, specific AUC values for key comparisons and p-values for the most important findings should be included to give readers immediate access to the study's primary quantitative outcomes.

Answer 1. We have taken your recommendation into account. We improved the ResultsÌ› s section: ”In our study, serum OSM was significantly associated with all four outcome measures: higher OSM levels predicted higher pMS (β = 0.471, p < 0.001, R² = 0.222), MES (β = 0.422, p < 0.001, R² = 0.178), NHI (β = 0.422, p < 0.001, R² = 0.256), and FC (β = 0.431, p < 0.001, R² = 0.186). Furthermore, ROC curve analyses demonstrated that OSM had excellent diagnostic accuracy for active disease, particularly in relation to histo-logical inflammation (AUC = 0.967). In comparison, FC showed good but slightly lower accuracy (AUC = 0.875). Notably, OSM also outperformed FC in discriminating histo-logical remission. Pairwise ROC curve analyses using DeLong’s test further confirmed the diagnostic accuracy of OSM, FC, and combined biomarker scores (OSM+FC) across clinical, endoscopic, and histological endpoints. The combined score PRE1 (OSM + FC based on NHI) achieved perfect discrimination (AUC = 1.000, p <0.001). Composite models PRE2 and PRE3 (OSM+FC based on MES and pMS) improved diagnostic accuracy relative to OSM, confirming the value of combining OSM with FC. Although both outperformed OSM (p < 0.05), neither achieved a superior advantage over FC.”

Q2. The manuscript needs a clear, conventional structure (Introduction, Methods, Results, Discussion, Conclusion) that facilitates reader comprehension and meets journal standards.

Answer 2. We are afraid that this structure cannot be changed. The article was originally written following the format (Introduction, Methods, Results, Discussion, Conclusion), but it was subsequently revised to its current form in accordance with the journal’s required standards.

Q3 + Q5. The Results section contains extensive statistical detail that, while thorough, may overwhelm readers. Consider moving some technical statistical details to supplementary materials and focusing the main text on the most clinically relevant findings.

While figures are referenced appropriately, the text could better integrate visual data with narrative interpretation. Each figure should be discussed in more detail regarding its clinical significance.

Answers 3+5. Thank you for your constructive feedback. I have already worked to minimize the statistical details in the main Results section, retaining only those necessary for clarity and interpretation. Additionally, I have reorganized the figures and incorporated them within the text, as advised, to strengthen the integration between the visual data and the accompanying narrative. I trust that the updated version adequately addresses your concerns and presents the findings with improved clarity and alignment with the journal’s standards.

Q4. The Discussion section is relatively brief, given the complexity of the findings. It would benefit from expansion to include more detailed interpretation of the clinical implications, comparison with specific prior studies, and discussion of mechanistic insights.

Answer 4. Thank you for recommendation. We have expanded the Discussion section by adding a dedicated subsection focused on the clinical implications of our findings. While several studies in the literature have investigated OSM, most do not address endoscopic and histological activity—the relevant data emphasized in our study. The majority of previous research has focused primarily on treatment response or diagnostic disease, which is outside the scope of our current work. The lack of research on OSM as a marker of histological activity underscores its potential utility in assessing mucosal inflammation in IBD and highlights the need for further studies on larger patient cohorts to validate these findings.

Q6. Phrases like "extensive or severe disease" (line 312) conflate disease extent with disease severity, which are distinct concepts in UC. Disease extent refers to anatomical involvement (E1-E3), while severity refers to inflammatory activity. These should be distinguished clearly.

Answer 6. In accordance with your recommendation, for a clearer statement I have revised the conclusion: ”Overall, our findings suggest that serum OSM correlates strongly with UC disease extent and severity, with markedly elevated concentrations in patients with extensive disease, as well as in those with severe disease activity.”

Q7. There is no discussion of missing data or how it was handled. Were all 89 patients included in every analysis? If data were missing, what approach was used?

Answer 7. Regarding the handling of missing data, as described in the Methods section, the inclusion and exclusion criteria were applied to define the study population. All 89 patients in the cohort met these criteria and were included in each analysis. No data were missing for the variables reported.

Q8. Discuss practical applications, cost-effectiveness considerations, and how OSM would integrate into current clinical practice and treatment algorithms.

Answer 8. The Discussions section has been improved and the following part has been added:

”3.3. Clinical Implications

Our data support incorporating histologic evaluation alongside clinic and endo-scopic assessment to achieve deeper remission in UC. The findings of our study suggest that serum OSM could serve as a valuable biomarker in the personalized management of UC. OSM measurement may help identify patients with histologic active disease, highlighting its role as a biomarker of mucosal inflammation, monitoring histological activity, and guide treatment decisions. Its strong correlation with microscopic in-flammation positions OSM as a potential tool for assessing histological healing, which is increasingly recognized as a key treatment target in UC management. Also, its abil-ity to more accurately identify patients with persistent histological inflammation could optimize therapeutic decision-making, potentially reducing unnecessary escalation of therapy or ineffective treatments. In particular, OSM could serve as a non-invasive biomarker to help identify patients with persistent microscopic inflammation, poten-tially reducing the frequency of endoscopic assessments in treat-to-target strategies.

A comparative evaluation of biomarkers against histological and endoscopic ref-erence standards provides essential guidance for optimizing daily clinical practice. Traditional systemic markers such as CRP and fibrinogen are nonspecific markers of inflammation but lack the precision needed to accurately reflect mucosal activity or confirm deep healing. In contrast, biomarkers that more directly represent local tissue inflammation—particularly OSM—demonstrate superior correlations with histologi-cal activity and histologic healing, making it valuable for identifying persistent micro-scopic disease. FC remains a strong noninvasive indicator of mucosal inflammation, but OSM stands out for its ability to detect subclinical, treatment-resistant inflamma-tion and to better predict true tissue-level remission. Integrating these more sensitive biomarkers into routine assessment, alongside clinical evaluation and endoscopic findings, supports a more accurate and individualized approach to disease manage-ment..

3.4. Cost-Effectiveness Considerations

Regarding cost-effectiveness and feasibility, while OSM measurement would in-volve additional laboratory costs compared to established markers such as FC, its abil-ity to more accurately reflect mucosal inflammation could optimize therapy selection and reduce unnecessary escalation, potentially offsetting costs over time. By guiding more precise interventions, OSM testing could contribute to improved long-term out-comes and cost savings within UC management pathways.”

Reviewer 3 Report

Comments and Suggestions for Authors

The manuscript titled “Serum Oncostatin M in Ulcerative Colitis Patients and Its Relation to Disease Activity” investigates the role of serum Oncostatin M (OSM) as a biomarker for ulcerative colitis (UC) activity across clinical, endoscopic, and histological domains. The study is well-structured and addresses a clinically relevant question, offering promising insights into non-invasive monitoring strategies.

Major Comments

  1. Study Design and Cohort

    • The sample size (n=89) is modest and single-center, which limits generalizability. Please provide a power calculation or justification for the chosen sample size.
    • Absence of a healthy control group or patients with other inflammatory conditions makes it difficult to assess the specificity of OSM for UC. Consider discussing this limitation more explicitly.
  2. Statistical Analysis

    • The ROC analysis is robust; however, the perfect AUC (1.000) for PRE1 suggests overfitting. Authors should clarify whether cross-validation or bootstrapping was performed to mitigate this risk.
    • Provide more detail on how cut-off values (e.g., OSM >21.75 pg/mL) were derived and discuss their reproducibility across different ELISA platforms.
  3. Clinical Applicability

    • While OSM shows strong correlation with histological activity, the manuscript should elaborate on how these findings could influence clinical decision-making. For example, could OSM replace or reduce the need for endoscopy in treat-to-target strategies?
    • Discuss the cost-effectiveness and feasibility of implementing OSM testing in routine practice compared to existing biomarkers like FC.
  4. Confounding Factors

    • The study does not address potential confounders such as medication use, disease duration, or comorbidities that might influence OSM levels. Please clarify whether these were controlled for or considered in the analysis.

Minor Comments

  1. Language and Style  
    • Units and consistency

      • Verify units: CRP typically mg/L (not mg/dL) in clinical practice; confirm and harmonize units across tables/text.
      • FC labeled “mcg/g”; prefer “µg/g” with consistent SI formatting.

    • Tables and descriptive statistics

      • For skewed variables (OSM, FC), provide median (IQR). Consider adding distribution plots (violin/boxplots) on a log scale.
    • Add abbreviations below all tables

Author Response

Dear Reviewer,

Thank you for your valuable time, for the positive feedback, and useful contribution on the manuscript ijms-4017391 entitled “Serum Oncostatin M in Ulcerative Colitis Patients and Its Relation to Disease Activity”. Your suggestions and recommendations have been taken into consideration. We much appreciate their input, which helped improve our manuscript.

Please find below a detailed point-by-point reply to the comments.

Sincerely,

--

Alina-Ecaterina Jucan, MD, PhD

Grigore T. Popa University of Medicine and Pharmacy Iasi, Romania

Institute of Gastroenterology and Hepatology Iasi, Romania

Q1. Study Design and Cohort

The sample size (n=89) is modest and single-center, which limits generalizability. Please provide a power calculation or justification for the chosen sample size.

Absence of a healthy control group or patients with other inflammatory conditions makes it difficult to assess the specificity of OSM for UC. Consider discussing this limitation more explicitly.

Answer 1. We acknowledge that the sample size (n = 89) is modest and from a single center, which may limit generalizability. Although formal power calculations were not performed, the cohort was defined using strict inclusion and exclusion criteria to ensure a well-characterized population, and statistically significant associations were observed despite the sample size.

We did not include a healthy control group or patients with other inflammatory conditions because the study focused on correlations between serum OSM with clinical, biochimical, endoscopic, and histological disease activity in UC. Serum OSM it’s a potential marker of disease extension and severity and correlates well with histological inflammation.

We recognize that this limits the assessment of OSM specificity, but future studies are planned to include comparative cohorts and investigate the role of OSM in predicting treatment response and diagnostic accuracy.

We added this statement in the Discussion section for a more explicit reason: ”Despite these strengths, several limitations should be acknowledged. This study did not include a healthy control group or patients with other inflammatory conditions, as it focused on correlations between serum OSM and clinical, biochemical, endoscopic, and histological disease activity in UC. The study was not designed to evaluate diagnostic performance or treatment response. Subsequent studies are needed to assess OSM in comparative populations and to explore its utility for both treatment monitoring and diagnostic purposes. ”

Q2. Statistical Analysis

The ROC analysis is robust; however, the perfect AUC (1.000) for PRE1 suggests overfitting. Authors should clarify whether cross-validation or bootstrapping was performed to mitigate this risk.

Provide more detail on how cut-off values (e.g., OSM >21.75 pg/mL) were derived and discuss their reproducibility across different ELISA platforms.

Answer 2. We appreciate these important comments. We acknowledge that the perfect AUC of 1.000 for PRE1 likely reflects model overfitting. No cross-validation or bootstrapping was performed in this exploratory analysis, and we have added a statement in the revised manuscript emphasizing that these results require validation in independent cohorts.

Regarding the cut-off values (e.g., OSM >21.75 pg/mL), these were determined using the Youden index from the ROC analysis to maximize sensitivity and specificity within our cohort. We recognize that absolute cut-off values may vary between ELISA platforms and laboratories; therefore, these thresholds should be interpreted cautiously. We have added a discussion highlighting the need for standardization of OSM measurement and external validation to assess reproducibility across different assay platforms.

Q3. Clinical Applicability

While OSM shows strong correlation with histological activity, the manuscript should elaborate on how these findings could influence clinical decision-making. For example, could OSM replace or reduce the need for endoscopy in treat-to-target strategies?

Discuss the cost-effectiveness and feasibility of implementing OSM testing in routine practice compared to existing biomarkers like FC.

Answer 3. The Discussions section has been improved and the following part has been added:

”3.3. Clinical Implications and Cost-Effectiveness Considerations

Our data support incorporating histologic evaluation alongside clinic and endoscopic assessment to achieve deeper remission in UC. The findings of our study suggest that serum OSM could serve as a valuable biomarker in the personalized management of UC. OSM measurement may help identify patients with histologic active disease, highlighting its role as a biomarker of mucosal inflammation, monitoring histological activity, and guide treatment decisions. Its strong correlation with microscopic inflammation positions OSM as a potential tool for assessing histological healing, which is increasingly recognized as a key treatment target in UC management. Also, its ability to more accurately identify patients with persistent histological inflammation could optimize therapeutic decision-making, potentially reducing unnecessary escalation of therapy or ineffective treatments. In particular, OSM could serve as a non-invasive biomarker to help identify patients with persistent microscopic inflammation, potentially reducing the frequency of endoscopic assessments in treat-to-target strategies.

Regarding cost-effectiveness and feasibility, while OSM measurement would involve additional laboratory costs compared to established markers such as FC, its ability to more accurately reflect mucosal inflammation could optimize therapy selection and reduce unnecessary escalation, potentially offsetting costs over time. By guiding more precise interventions, OSM testing could contribute to improved long-term outcomes and cost savings within UC management pathways.

 OSM could be integrated into existing clinical practice alongside established biomarkers and endoscopic assessments. Combined with clinical scoring systems and fecal biomarkers, OSM may help clinicians identify patients who require intensification of therapy or closer follow-up. Validation in larger, independent cohorts will be important before broad clinical implementation, but our findings provide a strong rationale for its inclusion in future treatment algorithms.”

Q4. Confounding Factors

The study does not address potential confounders such as medication use, disease duration, or comorbidities that might influence OSM levels. Please clarify whether these were controlled for or considered in the analysis.

Answer 4. Thank you for this remark. We acknowledge that potential confounders such as medication use, disease duration, and comorbidities were not controlled for in the current analysis. These factors were not included in the study design, and we have added a statement in the revised manuscript acknowledging this limitation. Future studies are planned to expand the analysis and assess the impact of these variables on serum OSM levels.

Minor Comments

Language and Style. Units and consistency

Verify units: CRP typically mg/L (not mg/dL) in clinical practice; confirm and harmonize units across tables/text.

FC labeled “mcg/g”; prefer “µg/g” with consistent SI formatting.

Tables and descriptive statistics

For skewed variables (OSM, FC), provide median (IQR). Consider adding distribution plots (violin/boxplots) on a log scale.

Add abbreviations below all tables

Answer.  All minor revisions have been addressed. Thank you!

Round 2

Reviewer 3 Report

Comments and Suggestions for Authors

accept ..All changes were done